# Triple Collocation of Ground-, Satellite- and Land Surface Model-Based Surface Soil Moisture Products in Oklahoma—Part I: Individual Product Assessment

**Zhen Hong [1], Hernan A. Moreno [2,3,*], Zhi Li [4], Shuo Li [4], John S. Greene [1], Yang Hong [4] and Laura V. Alvarez [2,3]**

[1] Department of Geography and Environmental Sustainability, University of Oklahoma, Norman, OK 73019, USA
[2] Department of Earth, Environmental and Resource Sciences, University of Texas, El Paso, TX 79902, USA
[3] NOAA-Cooperative Science Center for Earth System Sciences and Remote Sensing Technologies, New York, NY 10031, USA
[4] School of Civil Engineering and Environmental Science, University of Oklahoma, Norman, OK 73019, USA
[*] Correspondence: moreno@utep.edu

**Abstract:** Improvements in soil moisture observations and modeling play a vital role in drought, water resources, flooding, and landslide management and forecasting. However, the lack of multisensor products that integrate different spatial scales (i.e., from $1\ m^2$ to $10^2\ km^2$) is a pressing need in the management and forecasting chain. Up to date, surface soil moisture estimates could be obtained through three primary approaches: (1) in situ measurements and their interpolations, (2) remote sensing observations, and (3) land surface model (LSM) outputs. Each source of soil moisture has its own spatiotemporal resolution, strengths, and weaknesses. Therefore, their correct interpretation and application require an in-depth understanding of their accuracy and appropriateness. In this study, we explore the utility of the triple collocation (TC) method for an independent assessment of three soil moisture products to characterize their uncertainty structures and make recommendations toward a potential product merge. The state of Oklahoma is an ideal domain to test the hypotheses of this work because of the presence of marked west-to-east gradients in climate, vegetation, and soils. The three target soil moisture products include (1) the remotely sensed microwave soil moisture active passive (SMAP) L3_SM_P_E (9 km, daily), (2) the physically based LSM estimates from NLDAS_NOAH0125_H (1/8°, hourly; Noah), and (3) the Oklahoma Mesonet ground sensor network (point, 30 min). The product assessment was conducted from April 2015 to July 2019. The results indicate that, in general, Mesonet and Noah are the most reliable products, although their performance varies geographically and by land cover type, reflecting the main spatiotemporal characteristics and scope of each product. Specifically, Mesonet provides the best estimates of volumetric soil moisture with a mean Pearson correlation coefficient of 0.805, followed by Noah with 0.747. However, Noah represents the true soil moisture variation better than the interpolated Mesonet product on the mesoscale, with an averaged RMSE of 0.026 $m^3/m^3$. Over different land cover types, Mesonet had the best performance in shrub/scrub, herbaceous, hay/pasture, and cultivated crops with an average correlation coefficient of 0.79, while Noah achieved the best performance in evergreen, mixed, and deciduous forests, with an average correlation coefficient of 0.74. The period-integrated TC intercomparison results over nine climate divisions indicated that Noah outperformed in the central, northeast, and east-central regions. TC provides not only a new perspective for comparatively assessing multisource soil moisture products but also a basis for objective data merging to capitalize on the strengths of multisensor, multiplatform soil moisture products.

**Keywords:** soil moisture; triple collocation; SMAP; Mesonet; NLDAS; intercomparison

## 1. Introduction

　　Surface soil moisture ($\theta_s$) refers to the water held in the space between soil particles within the first few centimeters of the surface soil. This variable plays a fundamental role across spatial scales. At the plot and hillslope scales, it drives deeper-soil infiltration rates, runoff generation type and flux rate, soil evaporation, shallow-root plant transpiration, and surface energy flux partitioning [1] among other processes. At the regional scale, it is a fundamental factor in sustaining and ending droughts but also in triggering or enhancing floods and mass movements [1]. While soil moisture only accounts for a very small fraction (approximately 0.05%) of the total quantity of water within the global hydrological cycle, its uneven distribution (in space and time) plays a critical role in the climate and hydrologic systems [1,2]. Society depends on accurate measurements of soil moisture. Its correct estimation benefits precision agriculture through precise irrigation and fertilization [3,4]. At broader spatial scales, it enhances runoff and flood forecasting [5,6], drought monitoring and prediction [7,8], numerical weather forecasts [9–12], landslides [13,14], and wildfire predictions [15–17].

　　Currently, soil water content estimates can be obtained through three primary approaches: (1) in situ measurements, (2) remote sensing retrievals, and (3) land surface model (LSM) outputs. In situ soil moisture measurements have the ability to provide high, spatial, and temporal resolution of soil moisture at different depths [18–20]. There are several regional in situ networks designed for monitoring soil moisture within the United States, including the various state Mesonets, the Atmospheric Radiation Measurement Southern Great Plains (ARM-SGP), and the Soil Climate Analysis Network (SCAN) [20]. Within the state of Oklahoma, in addition to the state-wide network, finer-scale networks, such as the U.S. Department of Agriculture Agricultural Research Service Little Washita and Fort Cobb networks, are designed to have a higher density of stations over a smaller spatial domain [20]. Furthermore, field campaign activities, including the Southern Great Plains (SGP) hydrology experiments in 1997 and 1999, are sources of short-term, multiscale soil moisture measurements [18]. Despite all these efforts in building field-, regional- and national-scale soil moisture networks, the number of stations and their spatial coverage are still very limited by their inability to provide spatial representativeness of neighboring areas due to the high spatial heterogeneity of soil moisture [18,19,21]. One alternative to resolving the issue of complex spatial variability is the use of geostatistical techniques to interpolate (or extrapolate) in situ soil moisture measurements to neighboring areas. Nonetheless, results are often inaccurate when the spatial interpolations rely only on distance-related covariance functions [8,21–24] which is usually the case.

　　Land surface models, on the other hand, can provide soil moisture estimates at various depths with fixed spatiotemporal resolution. Spatially, resolutions (usually ranging from 1 to $10^2$ km$^2$ pixel size) are appropriate, but their model result quality is conditioned by the limited spatial resolution of the forcing inputs (e.g., remotely sensed precipitation fields of 1 km pixel size, hindering sub 1 km$^2$ variability) [25]. Finally, remote sensing-based soil moisture products from various orbital sensors working on different spectral bands (e.g., microwave, thermal, and optical) provide global-scale soil water content measurements within the first 1–10 cm of soil depth (i.e., $\theta_s$) [25], commonly with 1–100 km$^2$ spatial resolution. Among them, microwave remote sensing techniques have gained momentum over the past 20 years with their advantages in the fast and extensive retrieval of $\theta_s$ [26]. However, all microwave remote sensing soil moisture measurements using C, X, and L bands only measure soil moisture in the top five cm (or less) of the soil under low to moderate vegetation cover [27]. In summary, each source of soil moisture observations has its strengths and weaknesses. However, none of them, at least by themselves, are adequate for providing accurate $\theta_s$ data since their performance differs across diverse spatiotemporal scales and landcover types. Therefore, it is novel and useful to combine these (or, in the future, other) three (or more) independent data sources to capitalize on their individual strengths across scales and land surface types.

Traditionally, $\theta_s$ evaluations are carried out through direct comparisons of satellite or LSM outputs against point ground observations. However, the accuracy of such comparisons is often limited by the spatial representativity of each (i.e., point versus pixel)) [22]. Therefore, the metrics obtained from such comparisons may not truly reflect the error characteristics of the target soil moisture product. In response to this challenge, Scipal et al. [9] first proposed to use the triple collocation (TC) error estimation technique in soil moisture applications. TC analysis is a method for estimating the random error variances of three spatially and temporally collocated measurement systems of the same geophysical variable without treating any one system as perfectly observed "truth". Using the same assumptions as TC, McColl, et al. [28] developed the extended triple collocation (ETC), which provides the Pearson correlation coefficient as an additional validation metric to the root mean square error. The ETC has been widely used in the validation of satellite-based soil moisture retrievals in recent years. For example, Chen et al. [29] applied ETC-based validation techniques to the soil moisture active/passive (SMAP) Level 2 soil moisture product at five SMAP core validation sites and obtained an unbiased estimation of the satellite-versus-truth correlation metric. Chen et al. [30] adopted the ETC and conducted a global-scale assessment and inter-comparison of the SMAP Level 3, soil moisture ocean salinity (SMOS) Level 3, and advanced SCATterometer (ASCAT) Level 2 soil moisture products. Wu et al. [31] presented an ETC-based comprehensive assessment of SMAP, European Space Agency (ESA) Climate Change Initiative (CCI) Soil Moisture, and SMOS with in situ measurements in China. Xu et al. [32] conducted a global scale ETC-based evaluation of eight root zone soil moisture products, including GLDAS Noah, ERA-5, MERRA-2, NCEP R1, NCEP R2, JRA-55, SMAP level 4, and SMOS level 4 datasets.

Due to the characteristics of the above-mentioned three main sources of soil moisture measurement (in situ, land surface model, and satellite), their data quality and representativeness vary over different land cover types. For example, the Oklahoma Mesonet site standards minimize the influence of urban landscapes, irrigation, forests, bare soil, fast-growing vegetation, and large bodies of water [33]. It is suggested that vegetation at the site should be uniform and low growing, such as short grasses [34]. Therefore, soil moisture measurements at the Oklahoma Mesonet sites may not well represent SM variations over bare soil, crops, forests, and other fast-growing vegetation. On the other hand, the vegetation classification of the NLDAS land surface model was derived from the global, 1 km, AVHRR-based, 13-class vegetation database of the UMD (Noah; [35]). For each 1/8° grid cell, Noah uses the most predominant vegetation class [36]. Xia et al. [37] evaluated 20 years (January 1985–December 2004) of NLDAS-2 model-simulated soil moisture with in situ measurements over the continental United States and concluded that the performance for all models was better in the Southeast, Great Plains, Midwest, and Northwest, and lower in the Southwest and the Northeast with their dominant vegetation cover as forest, grassland, a mixture of cropland and grasslands, grassland, open shrubland, and forest, respectively. Zhang et al. [38] conducted a comprehensive validation of the SMAP Level 3 SM product with ground measurements over varied climates and landscapes from 1 April 2015, to 31 March 2018. Results showed that SMAP level 3 SM products had better performance over grassland than over cropland. In summary, these three benchmark and popular soil moisture products (e.g., Mesonet, Noah, and SMAP) are subject to representation inadequacies over various geographic locations and land cover types, and a correct interpretation of their value requires an in-depth understanding of their scope. Therefore, there is a visible gap in the literature regarding independent evaluations of triplets (or more) of soil moisture products at the state or regional level to determine their value and, upon performance, explore the possibility of merging them into a better multi-source product that outperforms each one individually.

This manuscript is the first of a series of two with the overarching goal of cross-evaluating three $\theta_s$ products of different spatial resolutions, independently across various land cover types and climatic regions within the state of Oklahoma (U.S) to then capitalize on their value for a further multi-product merge. Specifically, this first article will conduct

a comprehensive assessment of the satellite SMAP_L3 (SMAP), land surface NOAH model (Noah), and Mesonet soil moisture (Mesonet) at daily and seasonal timescales using the triple collocation method. The results of this study are expected to provide a basis for objective data merging to capitalize on the strengths of multi-sensor multiplatform $\theta_s$ products. The rest of the article is organized as follows: Section 2 shows the details of the data and study area; Section 3 describes methods and data processing; Section 4 presents the results and analysis; Section 5 provides a discussion; and Section 6 offers some conclusions and suggestions for future work.

## 2. Data Sources

### 2.1. In Situ Soil Moisture Product: The Oklahoma Mesonet

The Oklahoma Mesonet is a world-class, statewide network of environmental monitoring stations that was established in January 1994. It measures atmospheric, hydrologic, and meteorological variables including temperature, humidity, solar radiation, wind speed and direction, and soil moisture to aid in operational weather forecasting and environmental research across the state [33]. With at least one station in each of Oklahoma's 77 counties, the Mesonet consists of 120 automated stations across the state. These measurements are packaged into observations every 5 min, transmitted to the Oklahoma Climatological Survey (OCS) at the University of Oklahoma (OU), where the observed data are processed and verified for their quality, and then made public. Soil moisture data are collected every 30 min and recorded locally at each site, including at the surface [39]. Since SMAP measurements are instantaneous, the Mesonet soil moisture measurements at local solar times are 6 a.m. and 6 p.m. ranging from 4 January 2015 to 7 January 2019 were used in this study. A total of 115 Mesonet sites were selected according to the data availability during the study time (see Figure 1). Since the data points are spread out across the state, interpolation was conducted using ordinary Kriging, as previously suggested by [21,40] and regularly released by Oklahoma Mesonet then regridded to 9 km.

### 2.2. Model-Based Soil Moisture Product: NLDAS_NOAH0125_H

The North American Land Data Assimilation System phase 2 (NLDAS-2) is an offline data assimilation system running four land surface models (Noah, SAC-SMA, VIC, and Mosaic) over the conterminous United States (CONUS), the southern part of Canada, and the northern portion of Mexico with a 1/8° latitude-longitude resolution [41]. The four land surface models represent different methodological approaches to land surface modeling. This study uses the simulated soil moisture from the NLDAS-2 Noah model. The Noah model is the land model of the NCEP (National Centers for Environmental Modeling Prediction) operational regional and global weather and climate models [42–44]. It provides hourly soil moisture fields at 1/8° grid from 1979 to the present. The Noah model has four soil layers: 0–10 cm, 10–40 cm, 40–100 cm, and 100–200 cm and simulates soil moisture in the middle of each soil layer (5, 25, 70, and 150 cm). In this study, only the top layer value (i.e., 5 cm) is used to represent $\theta_s$. Xia et al. [41] compared soil moisture estimates of four NLDAS-2 land surface models (Noah, Mosaic, SAC, VIC) with three in-situ soil moisture observation data sets in the United States (the Illinois Climate Network, the Oklahoma Mesonet network, and the Soil Climate Analysis Network) to find that Noah had the smallest mean absolute error (MAE = 0.036), root mean square error (RMSE = 0.04) and bias (Bias = −0.033) in the comparison with the Oklahoma Mesonet observations for absolute daily soil moisture at the top 10 cm soil layer in a six-year period (from 1 January 1997 to 31 December 2002). Therefore, the hourly soil moisture simulations of NLDAS-2 Noah model between the period of 1 April 2015 to 1 July 2019 were used in this study.

### 2.3. Satellite Soil Moisture Product: SMAP L3_SM_P_E

Launched in January 2015, SMAP is an orbiting observatory that estimates the amount of water in the top 0–10 cm of soil everywhere on Earth's land surface every two to three days. SMAP was designed to provide high-resolution soil moisture information with radar

(active) and radiometer (passive) sensors that operate at L-band frequencies. However, the radar instrument terminated its operation due to the failure of its power supply after three months of data collection. The SMAP radiometer has been operating flawlessly and in an extended operation phase since 2018 [27].

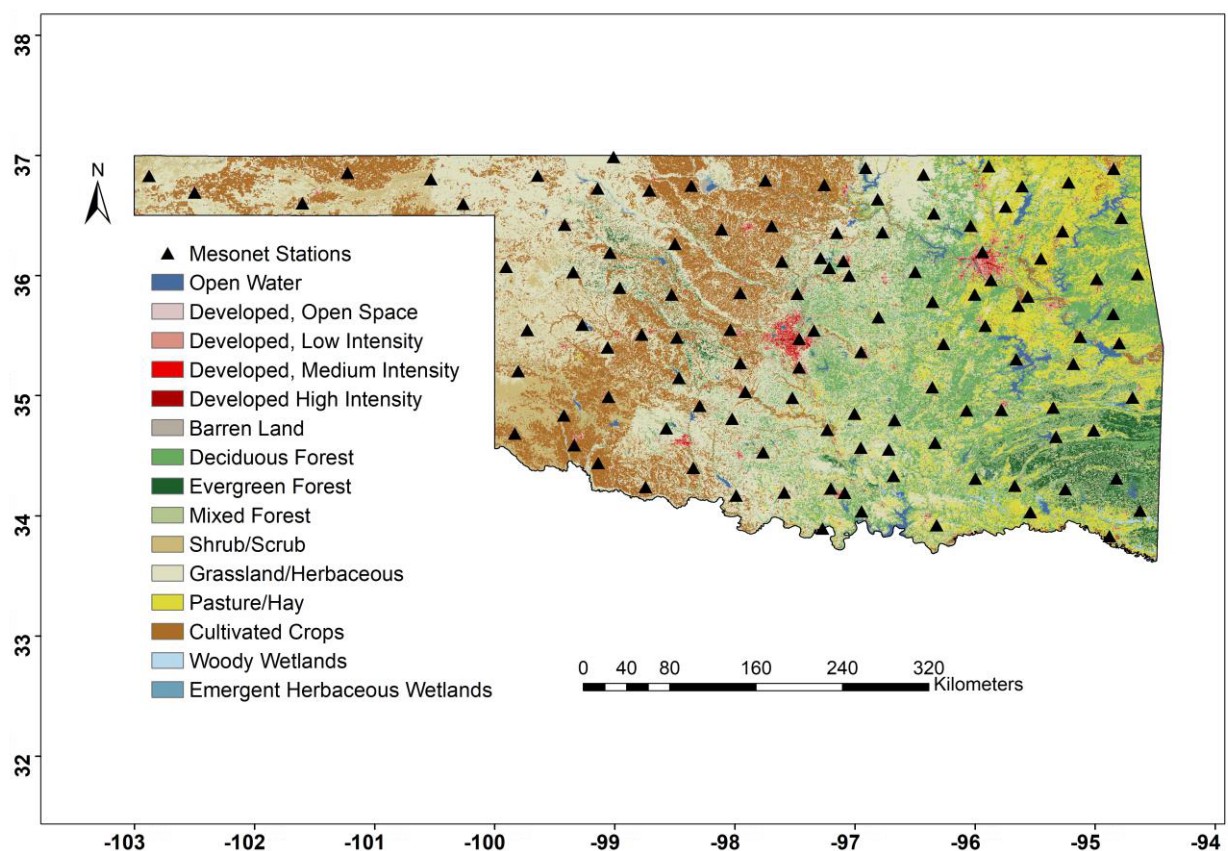

**Figure 1.** The distribution of in situ soil moisture stations from the Oklahoma Mesonet on a National Land Cover Dataset (NLCD) land cover type (for the year 2016) map. Geodetical coordinates (Lon, Lat) are also indicated.

In total, the SMAP mission has generated 23 distributable data products representing four levels of data processing. Level 1 products are instrument-related data sectioned into surface radar backscatter and brightness temperatures. Level 2 are geophysical retrievals in half orbit granules resulting from instrument data. Level 3, are daily global composites of Level 2 data for an entire UTC day and Level 4 products are outputs from geophysical models utilizing SMAP data [27]. The reasons why the SMAP Level 3 product is used in this study are: (1) Even though both Level 2 and Level 3 products are geophysical retrievals, Level 3 values are daily global composites of Level 2; and (2) While both Level 3 and Level 4 products are daily global retrievals, only Level 3 can satisfy the independency assumption of triple collocation analysis [9] since Level 4 are outputs from geophysical models that are not necessarily independent from the NLDAS2 Noah or Mesonet.

The SMAP Level 3 product used in this study is the Enhanced L3 Radiometer Global Daily 9 km EASEGrid Soil Moisture, Version 3 (L3_SM_P_E; [30]). It is a daily global composite of the enhanced SMAP L2_SM_P_E product, which contains gridded data of 6:00 a.m. (descending) and 6:00 p.m. (ascending) SMAP radiometer-based soil moisture retrievals, ancillary data, and quality assessment flags on the global 9-km Equal-Area Scalable Earth (EASE 2.0) grid. The main output of this dataset is 0–5 cm surface soil moisture, ($\theta_s$). This product is publicly available through the National Snow and Ice Data Center. Surface soil moisture data of SMAP L3_SM_P_E product pertaining to the period from 1 April 2015 to 1 July 2019 were used in this study [45,46].

## 2.4. Auxiliary Data

To better understand the performance of the three soil moisture products over different land cover types, the national land cover dataset (NLCD) 2016 product was used in this study. The NLCD provides nationwide data on continental U.S. land cover and land cover change at a 30 m resolution with a 16-class legend based on a modified Anderson Level II classification system. There are fifteen land cover types within the state of Oklahoma as shown in Figure 1. Oklahoma is divided into nine climate divisions (Figure 2). These nine divisions obey to multiple factors, such as climatic conditions, county lines, crop districts, and drainage basins rather than strict climatic homogeneity [47]. Therefore, analyzing the performance of three soil moisture products across the nine climate divisions also provides a unique opportunity to better understand how regional properties influence soil moisture estimation.

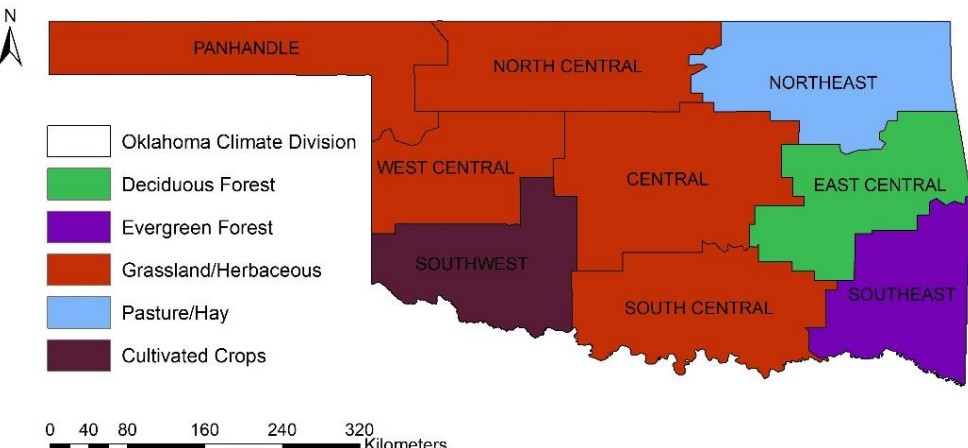

**Figure 2.** Oklahoma climate divisions with their major land cover types based on the 2016 NLCD map.

## 3. Data Processing and the Triple Collocation Method

### 3.1. Data Processing

Table 1 presents some information about the product triplet, including version, data spans, spatiotemporal resolutions, and soil depth to represent $\theta_s$.

**Table 1.** Summary metadata of satellite (SMAP L3_SM_P_E herein called SMAP), model (NLDAS_NOAH0125_H herein called Noah), and in situ (Oklahoma Mesonet) soil moisture products used in this study.

| Data | Version | Available Data Period | Temporal Resolution | Spatial Resolution | Depth |
|------|---------|----------------------|---------------------|--------------------|-------|
| SMAP | L3_SM_P_E | 2015—present | daily | 9 km | 0–5 cm |
| Noah | 0125_H | 1979—present | hourly | 0.125° | 0–10 cm |
| Mesonet | 115 sites | 1998—present | 30 min/daily | point | 0–5 cm |

Since the TC analysis requires three spatially and temporally collocated measurement systems, the grid of SMAP (EASE_v2) was defined as the reference for the three products. Therefore, Noah data were resampled to this grid using the area-weighted average method while Oklahoma Mesonet soil moisture measurements were matched to the EASE_v2 grid using ordinary kriging [21,40]. Figure 3 shows the temporal coverage of three soil moisture products during the study period (April 2015 through July 2019). Mesonet and Noah provide continuous data, but SMAP measurements are intermittent because the same swath from each orbit of SMAP is only repeated every eight days. The three collocated soil moisture products (SMAP, Mesonet, and Noah) are evaluated at daily and seasonal timescales using the TC method over different land cover types across Oklahoma. For the comparison, two-time stamps are selected, one diurnal (6 a.m. LST) and one nocturnal

(6 p.m. LST) according to the availability of SMAP. Table 2 shows the number of collocated samples for each grid point at daily and seasonal timescales. All sample sizes are larger than the suggested TC sample size (100) by Scipal et al. [9].

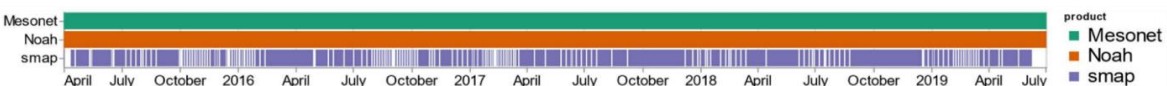

**Figure 3.** Temporal coverage of SMAP, Noah, and Mesonet soil moisture products used in this study. White spaces represent data gaps.

**Table 2.** Sample sizes of 6 a.m. and 6 p.m. Seasonal TC triplets used in this study.

| Type | Spring | Summer | Autumn | Winter |
|---|---|---|---|---|
| Number | 394 | 340 | 326 | 360 |

### 3.2. Classical Triple Collocation

Traditionally, one-to-one comparisons have been used as the main technique for verification of satellite and model products including soil moisture, precipitation, and evapotranspiration among other variables. However, the accuracy of such one-to-one comparisons between satellite or model outputs against in-situ measurements is often limited by differences in spatial and temporal representation (e.g., time-averaged versus instantaneous or point versus satellite footprint) [48]. Therefore, the metrics obtained from such comparisons may not truly reflect the nature of the error characteristics of each soil moisture product. In response to this challenge, Scipal et al. [9] first proposed to use the Triple collocation (TC) error estimation technique in soil moisture applications. TC analysis is a method for estimating the random error variances of three spatially and temporally collocated measurement systems of the same geophysical variable without treating any one system as perfectly observed "truth". A few assumptions are necessary for the TC method: (1) linearity between the true soil moisture signal and the observations, (2) signal and error stationarity, i.e., their mean values and variances are assumed to remain constant over time, (3) error orthogonality, i.e., the errors are independent of the true soil moisture signal, (4) the errors of three independent products should be independent or unrelated which means they must have a zero cross-correlation, and (5) the expectation of error is treated as zero. Yilmaz and Crow [49] conducted experiments on the TC errors due to the relevance of three products, and the results revealed that the more independent they are, the less TC-induced error there will be. It is essential to consider the relevance of the inputs to make the TC method more reliable [50]. The three selected soil moisture products, i.e., ground-based (Mesonet), model-based (Noah), and satellite-based (SMAP) all meet the above criteria.

The TC method treats all three independent products as equally important, and thus no preference or bias is introduced for any one approach. Equation (1) illustrates a standard form of the TC method [51]:

$$R_i = a_i + b_i T + \varepsilon_i \tag{1}$$

where, $R_i$ (i $\in$ ($X, Y, Z$)) indicates each of the three collocated soil moisture datasets X, Y, and Z, T is the "relative truth", $a_i$ and $b_i$ are the weights and biases to adjust, and $\varepsilon_i$ represents the error for each product *i*. Given this definition, the covariances between pairs of two different measurement systems (e.g., X and Y) would be given by

$$Cov(R_X, R_Y) = E(R_X R_Y) - E(R_X)E(R_Y) = b_X b_Y \sigma_T^2 + b_X Cov(T, \varepsilon_Y) + b_Y Cov(T, \varepsilon_X) + Cov(\varepsilon_X, \varepsilon_Y) \tag{2}$$

where, $\sigma_T^2 = Var(T)$, according to assumptions (3), (4), (5), $E(\varepsilon_X) = 0$, $Cov(\varepsilon_X, \varepsilon_Y) = 0$, $X \neq Y$, and $Cov(T, \varepsilon_X) = 0$. Therefore, Equation (2) reduces to

$$Q_{XY} = Cov(R_X, R_Y) = \begin{cases} b_X b_Y \sigma_T^2 & for\ X \neq Y \\ b_X^2 \sigma_T^2 + \sigma_{\varepsilon_X}^2 & for\ X = Y \end{cases} \tag{3}$$

where, $\sigma_{\varepsilon_X}^2 = Var(\varepsilon_X)$, since there are seven unknowns ($b_X, b_Y, b_Z, \sigma_{\varepsilon_X}, \sigma_{\varepsilon_Y}, \sigma_{\varepsilon_Z}, \sigma_T$) in six equations in the $3 \times 3$ covariance matrix ($Q_{XX}, Q_{XY}, Q_{XZ}, Q_{YY}, Q_{YZ}, Q_{ZZ}$), there is no unique solution. However, the introduction of a new variable $\theta_X = b_X \sigma_T$, changes (3) to

$$Q_{XY} = Cov(R_X, R_Y) = \begin{cases} \theta_X \theta_Y & for\ X \neq Y \\ \theta_X^2 + \sigma_{\varepsilon_X}^2 & for\ X = Y \end{cases} \tag{4}$$

From Equation (4), we now have six unknowns in six equations and are able to calculate the root mean square error (RMSE) in the set of Equation (5) that are based on the covariance of triplets [52]:

$$\sigma_\varepsilon = \begin{cases} \sqrt{Q_{XX} - \frac{Q_{XY}Q_{XZ}}{Q_{YZ}}} \\ \sqrt{Q_{YY} - \frac{Q_{XY}Q_{YZ}}{Q_{XZ}}} \\ \sqrt{Q_{ZZ} - \frac{Q_{XZ}Q_{YZ}}{Q_{XY}}} \end{cases} \tag{5}$$

### 3.3. Extended Triple Collocation

Using the same assumptions as TC, McColl et al. [28] introduced an additional performance metric, the Pearson correlation coefficient (CC) of the measurement system with respect to the unknown target with the called "ETC" method in which CC is calculated as a set of Equation (6).

$$\begin{cases} CC_X{}^2 = \frac{Q_{XY}Q_{XZ}}{Q_{XX}Q_{YZ}} \\ CC_Y{}^2 = \frac{Q_{XY}Q_{YZ}}{Q_{YY}Q_{XZ}} \\ CC_Z{}^2 = \frac{Q_{XZ}Q_{YZ}}{Q_{ZZ}Q_{XY}} \end{cases} \tag{6}$$

### 3.4. Use of the Classical and Extended Triple Collocation for the Three Testing Products

The mathematical derivations explained in equations 4 through 6 will be used to evaluate the three study products (i.e., SMAP, Noah, and Mesonet) considering the ETC method. Since both RMSE and CC are derived from covariances between the three products, they reveal the relative error as a measurement of their uncertainty. Therefore, the least uncertain product, represented by the lowest RMSE and highest CC, will have the best performance. Likewise, the most uncertain product will be associated with the highest RMSE, and lowest CC. Results are presented through gridded maps of instantaneous and seasonally-discretized (i.e., Spring, Summer, Fall, Winter) RMSE and CC and boxplots for groups of pixels with the same land cover type. The instantaneous values for comparison are extracted for two hours of the day, 6:00 a.m. and 6:00 p.m., determined by the available SMAP satellite geographical overpasses during the period of April 2015 to July 2019. The TC analysis is conducted over the entire state of Oklahoma and the different land cover types are extracted from the auxiliary data (see Section 2.4) to test the degree of dependency of each product's performance within different land cover types. According to the NLCD 2016 product, there are fifteen (15) land cover categories in the state (Figure 1). Table 3 contains the number of selected TC intercomparison pixels with a spatial resolution of nine (9) km for each land cover type in Oklahoma except open water, woody wetlands, and emergent herbaceous wetlands. To achieve statistical representativeness and preserve class diversity, the developed low intensity, medium intensity, and high intensity are classified as one land cover type "developed." Moreover, the triple product comparison was conducted for land cover types with more than 10 co-located pixels state-wide to guarantee some statistical significance of the results.

**Table 3.** Number of 9 km $\times$ 9 km grid cells for each of the land cover types in Oklahoma.

| Land Cover Type | Number of Co-Located Pixels |
|---|---|
| Developed Open Space | 137 |
| Developed Low Intensity | 25 |
| Developed Medium Intensity | 7 |

**Table 3.** *Cont.*

| Land Cover Type | Number of Co-Located Pixels |
|---|---|
| Developed High Intensity | 7 |
| Deciduous Forest | 414 |
| Evergreen Forest | 74 |
| Mixed Forest | 43 |
| Shrub/Scrub | 124 |
| Grassland/Herbaceous | 744 |
| Hay/Pasture | 229 |
| Cultivated Crops | 338 |
| Barren Land (Rock/Sand/Clay) | 7 |

## 4. Results

### 4.1. Product Intercomparison of Soil Moisture Values

Figure 4 shows both CC and RMSE obtained for simultaneous 6:00 a.m. observations over the entire state of Oklahoma after applying the ETC method to the three independent products. Overall, Mesonet provides the highest spatiotemporal integrated CC ($CC_{mean}$) of 0.805, followed by Noah with a $CC_{mean}$ of 0.747 while the SMAP results with the lowest $CC_{mean}$ of 0.314 are at the state level. However, both skill and error metrics vary across geographic locations and products (Figure 4, Table 4). Results are not presented for the 6 p.m. case, due to the high similarity of results to the 6 a.m. measurements. Based on the climate divisions from NOAA's Climate Divisional Database (Table 4), Mesonet provides high average CC values in the southwest, west-central, south-central, and panhandle (0.92, 0.9, 0.88, and 0.88, respectively) regions, while providing a lower averaged correlation in the northeast (CC = 0.58). Noah exhibits high averaged CC in the panhandle and south-central (CC = 0.79 and 0.86, respectively) regions but lower values in the northeast and west-central divisions (CC = 0.66 and 0.67, respectively). The regional mean CC values of SMAP that present a stripe pattern on the central and east divisions of the state are, generally, higher in the panhandle, southwest, and west-central divisions (CC = 0.49, 0.47, and 0.39, respectively) than those in the other six climate divisions.

**Table 4.** Average CC and RMSE values obtained from the TC triplets at local 6:00 a.m. for Mesonet, Noah, and SMAP for nine climate divisions in Oklahoma. Values are similar to the 6:00 p.m. case.

| Division | $RMSE_{mean}$ | | | $CC_{mean}$ | | |
|---|---|---|---|---|---|---|
| | Mesonet | Noah | SMAP | Mesonet | Noah | SMAP |
| Panhandle | 0.04 | 0.02 | 0.06 | 0.88 | 0.79 | 0.49 |
| West Central | 0.04 | 0.03 | 0.07 | 0.9 | 0.67 | 0.39 |
| Southwest | 0.04 | 0.02 | 0.06 | 0.92 | 0.75 | 0.47 |
| North Central | 0.06 | 0.03 | 0.11 | 0.8 | 0.7 | 0.32 |
| Central | 0.05 | 0.03 | 0.09 | 0.75 | 0.75 | 0.28 |
| South Central | 0.04 | 0.02 | 0.09 | 0.88 | 0.86 | 0.33 |
| Northeast | 0.09 | 0.04 | 0.18 | 0.58 | 0.66 | 0.12 |
| East Central | 0.06 | 0.02 | 0.15 | 0.76 | 0.76 | 0.23 |
| Southeast | 0.05 | 0.02 | 0.16 | 0.85 | 0.76 | 0.25 |

In terms of the RMSE, Noah provides the smallest state-wide $RMSE_{mean}$ of 0.026 m$^3$/m$^3$, followed by Mesonet ($RMSE_{mean} = 0.054$ m$^3$/m$^3$) and SMAP ($RMSE_{mean} = 0.107$ m$^3$/m$^3$), but the error varies across different locations and products (Figure 4, Table 4). Mesonet shows low RMSE values (0.04 m$^3$/m$^3$) in the southwest, south-central, west-central, and panhandle but high values in the northeast (0.09 m$^3$/m$^3$). Noah exhibits small RMSE mean values in all nine divisions (equal or less than 0.04 m$^3$/m$^3$). SMAP (that, analogous to the CC map, presents a stripe pattern across the central and east climate divisions) exhibits low mean RMSE values (0.06 m$^3$/m$^3$) in the panhandle and southwest but higher in northeast, east-central, and southeast (0.18, 0.15, and 0.16 m$^3$/m$^3$, respectively). In summary,

among the nine climate divisions, Noah has the best performance in the central (CC = 0.75, RMSE = 0.03 $m^3/m^3$), northeast (CC = 0.66, RMSE=0.04 $m^3/m^3$), and east-central division (CC = 0.76, RMSE = 0.02 $m^3/m^3$), followed by Mesonet. In the other six climate divisions, Mesonet has higher mean CC values than Noah, while Noah provides lower averaged RMSE values than Mesonet. SMAP consistently presents lower-than-Mesonet (or Noah) averaged CC and higher-than-Mesonet (or Noah) averaged RMSE values over all nine climate divisions. Summarizing the strengths of each product, Mesonet has better performance in the panhandle, southwest, west-central, and south-central and worse performance in the north-central and northeast. Noah has better skill scores in the panhandle and sout-central but worse in the north-central, northeast, and west-central. SMAP performs better in the panhandle, southwest, west-central, and south-central but has less value (according to the TC assessment) in the northeast, east-central, and southeast divisions. All three products exhibit their poorest performances in the Oklahoma Northeast region.

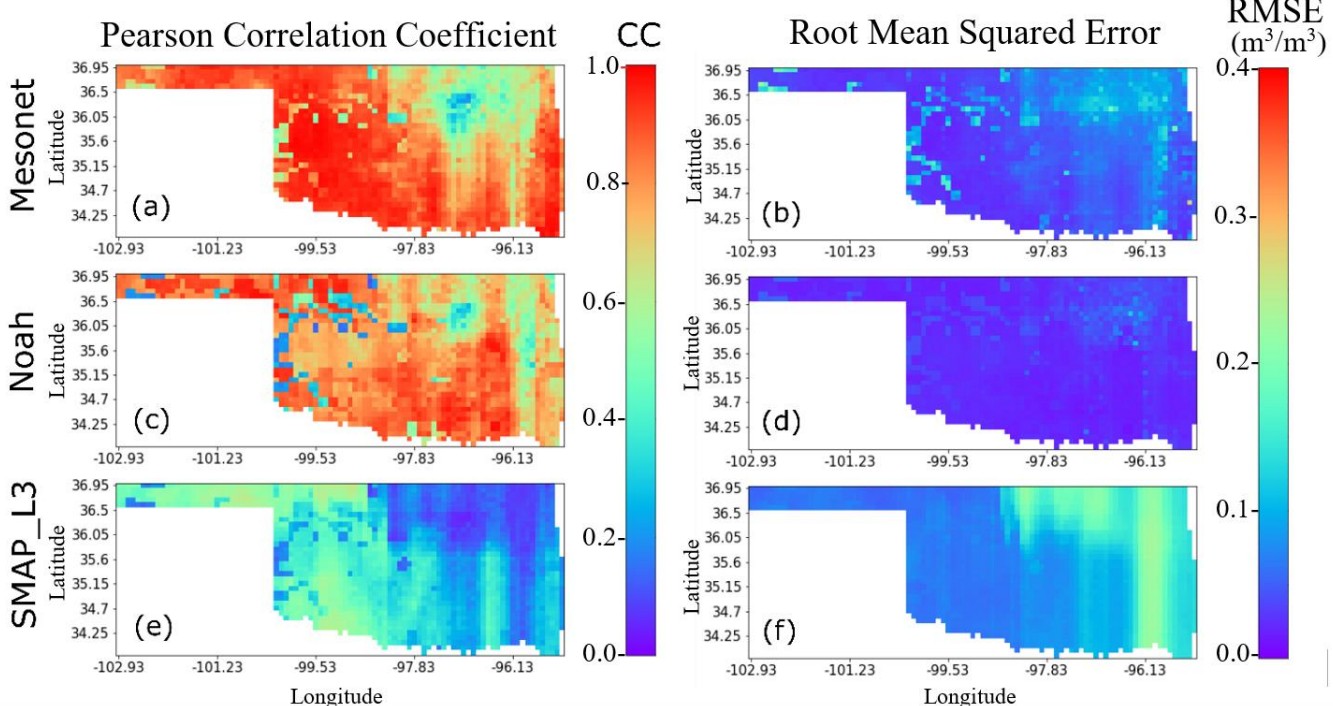

**Figure 4.** Product intercomparison assessment through both CC (left column) and RMSE (right column) after applying the ETC method for Mesonet (first row), Noah (second row), and SMAP (third row) on morning 6 a.m. $\theta_s$ values from April 2015 to July 2019. Geographical values are similar for the 6 p.m. case.

### 4.2. Seasonal Disaggregation of Soil Moisture Products Performance

The correlations and error patterns shown in Figure 4 are temporarily disaggregated in Figure 5 and Table 5 (Spring), Figure 6 and Table 6 (Summer), Figure 7 and Table 7 (Fall), and Figure 8 and Table 8 (Winter) to show the spatial distribution of seasonally averaged CC and RMSE values obtained from the TC triplets at local 6:00 a.m. over the entire state. The results gathered for local 6:00 p.m. are highly similar and consistent with those for 6:00 a.m. and the spatial patterns resemble those appearing in Figure 4. However, fall appears to be the season with the best performance of both Mesonet and Noah since the CC is higher across geographic regions. Nonetheless, winter and spring appear as the ones in which Mesonet and Noah have lower (than their annual average) skills across the state. Despite its lowest (among the three products) correlation coefficients, SMAP seems to have a higher value precisely during the winter and spring seasons across specific regions of the state in the north central, east central, and southeast sections of Oklahoma. Seasonally, products presented the following performance:

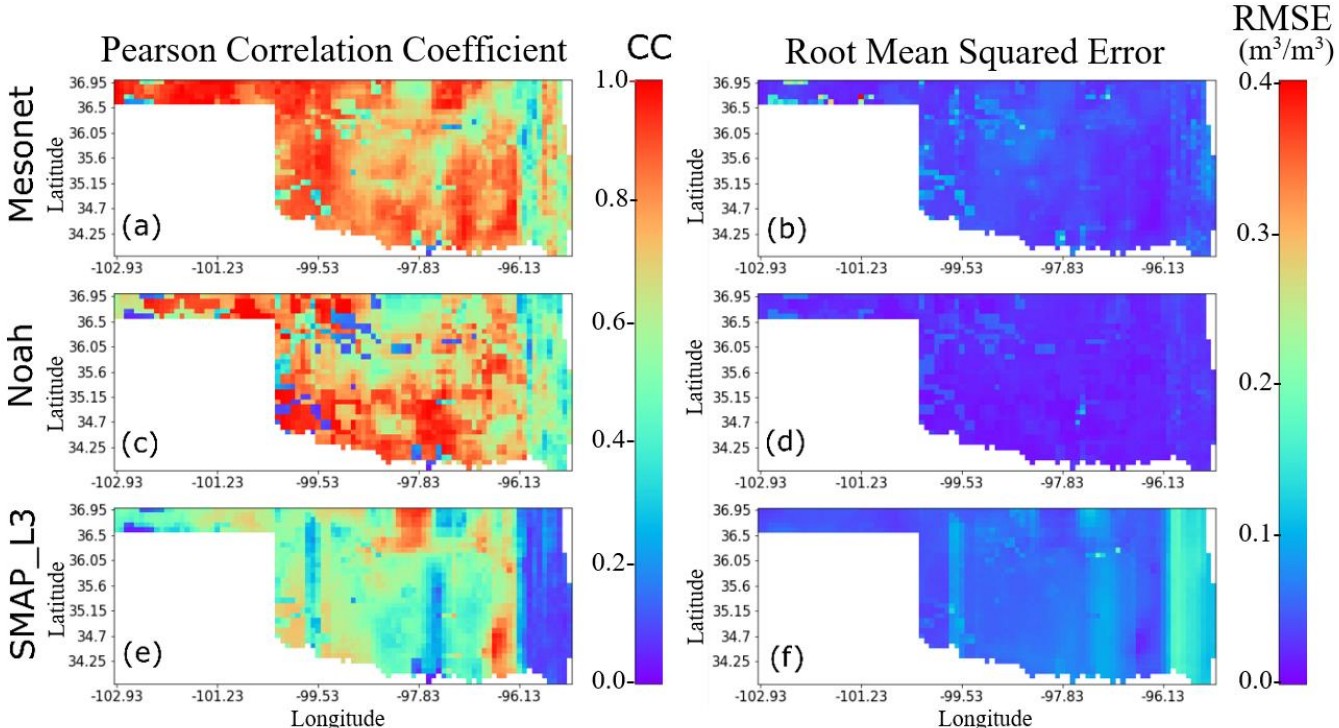

**Figure 5.** Product intercomparison assessment through CC (left column) and RMSE (right column) after applying the TC method for Mesonet (first row), Noah (second row), and SMAP (third row) surface soil moisture products based on morning 6 a.m. values from April 2015 to July 2019, integrated (averaged) only during the spring season months (i.e., March, April, and May).

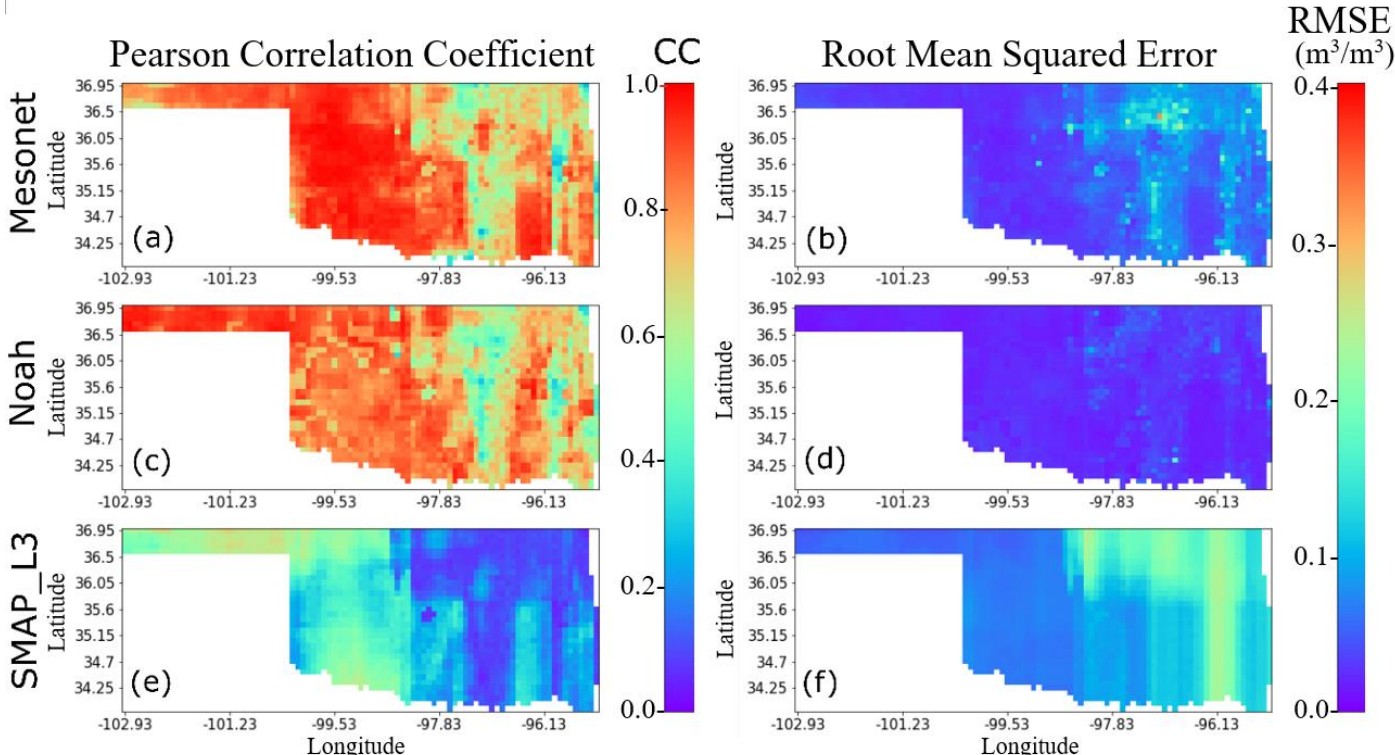

**Figure 6.** Product intercomparison assessment through CC (left column) and RMSE (right column) after applying the ETC method for Mesonet (first row), Noah (second row), and SMAP (third row) surface soil moisture products based on morning 6 a.m. values from April 2015 to July 2019, integrated (averaged) only during the summer season months (i.e., June, July, and August).

**Table 5.** Climate-division averaged CC and RMSE from the TC triplets at local 6:00 a.m. from April 2015 to July 2019, for Mesonet, Noah, and SMAP during the spring season months (i.e., March, April, and May) for nine climate divisions in Oklahoma.

| Division | $RMSE_{mean}$ | | | $CC_{mean}$ | | |
|---|---|---|---|---|---|---|
| | **Mesonet** | **Noah** | **SMAP** | **Mesonet** | **Noah** | **SMAP** |
| Panhandle | 0.04 | 0.02 | 0.04 | 0.89 | 0.75 | 0.51 |
| West Central | 0.05 | 0.03 | 0.05 | 0.82 | 0.66 | 0.54 |
| Southwest | 0.05 | 0.02 | 0.06 | 0.78 | 0.77 | 0.58 |
| North Central | 0.05 | 0.03 | 0.06 | 0.76 | 0.59 | 0.61 |
| Central | 0.04 | 0.03 | 0.06 | 0.71 | 0.7 | 0.47 |
| South Central | 0.03 | 0.02 | 0.07 | 0.81 | 0.79 | 0.43 |
| Northeast | 0.04 | 0.04 | 0.09 | 0.66 | 0.61 | 0.42 |
| East Central | 0.03 | 0.02 | 0.1 | 0.71 | 0.66 | 0.37 |
| Southeast | 0.04 | 0.02 | 0.11 | 0.65 | 0.57 | 0.22 |

**Table 6.** Climate-division averaged CC and RMSE from the TC triplets at local 6:00 a.m. from April 2015 to July 2019, for Mesonet, Noah, and SMAP during the summer season months (i.e., June, July, and August) over nine climate divisions in Oklahoma.

| Division | $RMSE_{mean}$ | | | $CC_{mean}$ | | |
|---|---|---|---|---|---|---|
| | **Mesonet** | **Noah** | **SMAP** | **Mesonet** | **Noah** | **SMAP** |
| Panhandle | 0.03 | 0.02 | 0.06 | 0.87 | 0.91 | 0.56 |
| West Central | 0.03 | 0.02 | 0.07 | 0.96 | 0.82 | 0.36 |
| Southwest | 0.03 | 0.02 | 0.07 | 0.93 | 0.83 | 0.45 |
| North Central | 0.05 | 0.02 | 0.11 | 0.83 | 0.82 | 0.31 |
| Central | 0.06 | 0.03 | 0.1 | 0.78 | 0.72 | 0.22 |
| South Central | 0.06 | 0.03 | 0.1 | 0.78 | 0.78 | 0.23 |
| Northeast | 0.1 | 0.03 | 0.19 | 0.68 | 0.65 | 0.1 |
| East Central | 0.07 | 0.02 | 0.15 | 0.71 | 0.7 | 0.17 |
| Southeast | 0.06 | 0.02 | 0.16 | 0.8 | 0.71 | 0.22 |

**Table 7.** Climate-division averaged CC and RMSE from the TC triplets at local 6:00 a.m. values from April 2015 to July 2019, for Mesonet, Noah, and SMAP during the fall season months (i.e., September, October, and November) for nine climate divisions in Oklahoma.

| Division | $RMSE_{mean}$ | | | $CC_{mean}$ | | |
|---|---|---|---|---|---|---|
| | **Mesonet** | **Noah** | **SMAP** | **Mesonet** | **Noah** | **SMAP** |
| Panhandle | 0.03 | 0.02 | 0.07 | 0.9 | 0.87 | 0.51 |
| West Central | 0.03 | 0.02 | 0.07 | 0.95 | 0.81 | 0.43 |
| Southwest | 0.02 | 0.02 | 0.06 | 0.98 | 0.81 | 0.5 |
| North Central | 0.05 | 0.02 | 0.12 | 0.79 | 0.8 | 0.32 |
| Central | 0.04 | 0.02 | 0.09 | 0.85 | 0.87 | 0.34 |
| South Central | 0.04 | 0.02 | 0.09 | 0.94 | 0.87 | 0.38 |
| Northeast | 0.08 | 0.03 | 0.19 | 0.68 | 0.73 | 0.14 |
| East Central | 0.06 | 0.02 | 0.15 | 0.81 | 0.83 | 0.21 |
| Southeast | 0.05 | 0.02 | 0.18 | 0.82 | 0.87 | 0.23 |

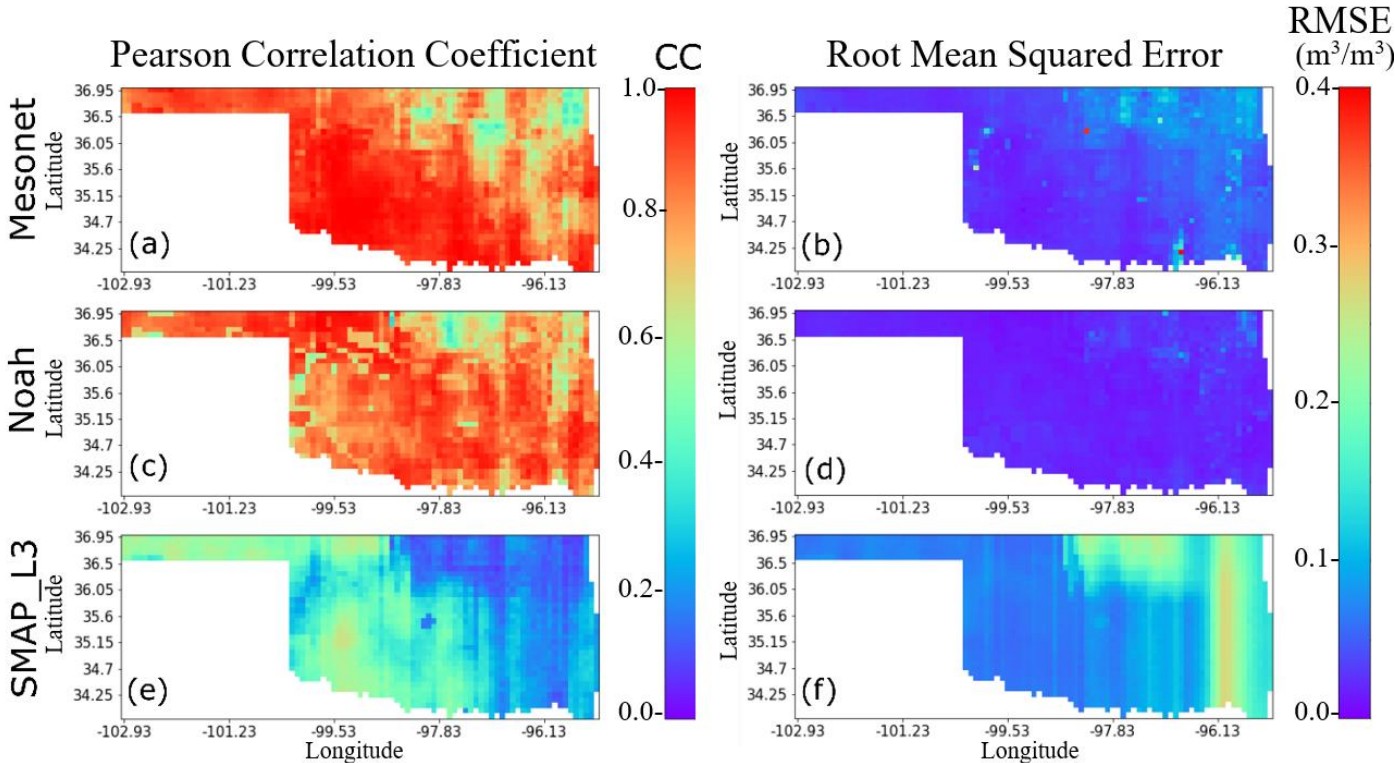

**Figure 7.** Product intercomparison assessment through CC (left column) and RMSE (right column) after applying the TC method for Mesonet (first row), Noah (second row), and SMAP (third row) surface soil moisture products based on morning 6 a.m. values from April 2015 to July 2019, integrated (averaged) only during the fall season months (i.e., September, October, and November).

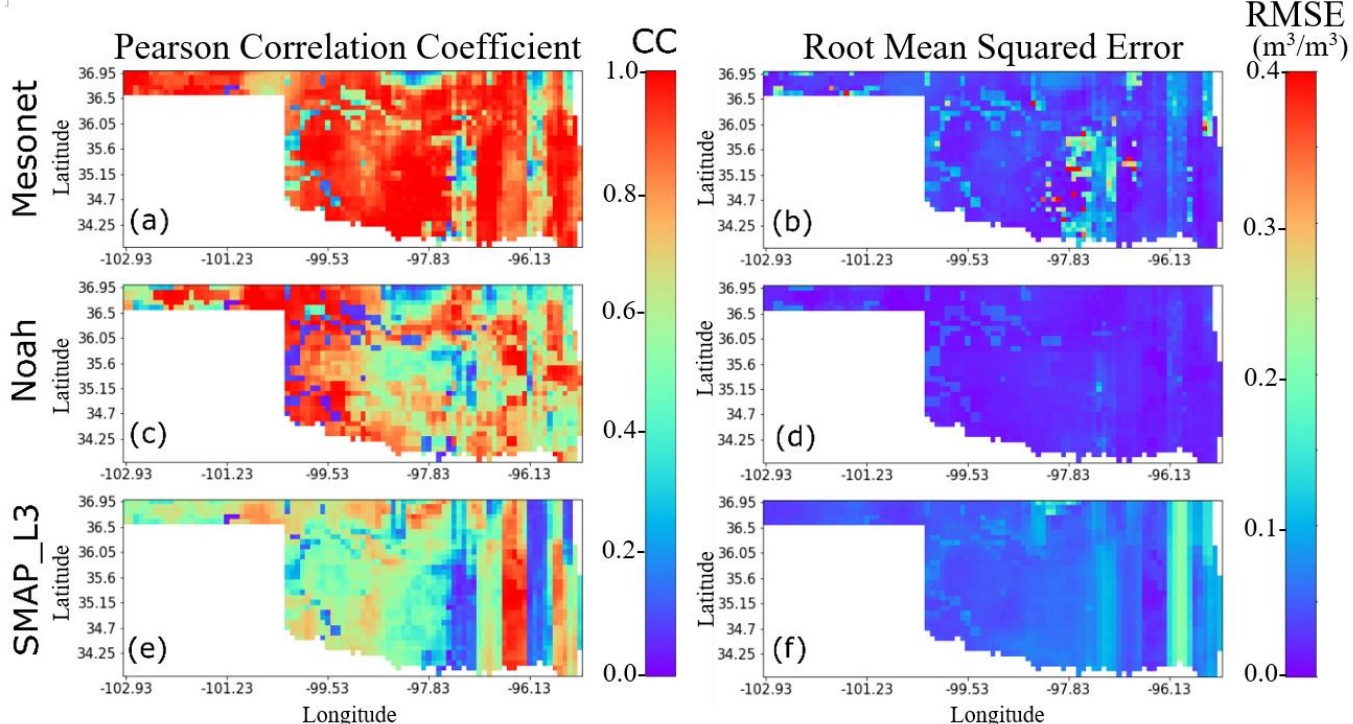

**Figure 8.** Product intercomparison assessment through CC (left column) and RMSE (right column) after applying the TC method for Mesonet (first row), Noah (second row), and SMAP (third row) surface soil moisture products based on morning 6 a.m. values from April 2015 to July 2019, integrated (averaged) only during the winter season months (i.e., December, January, and February).

**Table 8.** Climate-division averaged CC and RMSE from the TC triplets at local 6:00 a.m. values from April 2015 to July 2019, for Mesonet, Noah, and SMAP during the winter season months (i.e., December, January, and February) for nine climate divisions in Oklahoma.

| Division | RMSE$_{mean}$ | | | CC$_{mean}$ | | |
|---|---|---|---|---|---|---|
| | **Mesonet** | **Noah** | **SMAP** | **Mesonet** | **Noah** | **SMAP** |
| Panhandle | 0.06 | 0.02 | 0.04 | 0.84 | 0.78 | 0.59 |
| West Central | 0.04 | 0.02 | 0.05 | 0.86 | 0.7 | 0.52 |
| Southwest | 0.04 | 0.02 | 0.05 | 0.85 | 0.73 | 0.61 |
| North Central | 0.05 | 0.02 | 0.06 | 0.78 | 0.58 | 0.58 |
| Central | 0.08 | 0.03 | 0.06 | 0.84 | 0.55 | 0.43 |
| South Central | 0.07 | 0.03 | 0.07 | 0.89 | 0.68 | 0.4 |
| Northeast | 0.05 | 0.02 | 0.09 | 0.81 | 0.66 | 0.47 |
| East Central | 0.05 | 0.02 | 0.08 | 0.87 | 0.72 | 0.6 |
| Southeast | 0.03 | 0.02 | 0.1 | 0.85 | 0.66 | 0.55 |

During the spring season (Figure 5 and Table 5), among the nine climate divisions, Mesonet had higher mean CC values than Noah, while Noah provided lower averaged RMSE values than Mesonet. SMAP CC values were consistently lower-than-Mesonet (or Noah), and the RMSE was higher-than-Mesonet (or Noah) over all nine climate divisions. Regionally, Mesonet showed high average CC values in the Panhandle, West Central, and Southwest. However, the averaged RMSE values for this product in these divisions were higher than Noah's. Noah, on the other hand, illustrated better performance in the south central and southwest divisions, and was worse in the north-central and northeast climate divisions. Finally, SMAP performed best in the panhandle, southwest, west central, and north-central divisions but presented poorer results in the northeast, east-central, and southeast divisions.

During the summer season (Figure 6 and Table 6), Mesonet has better performance in the West Central, Southwest, and Panhandle, but a poorer performance in the Northeast, East Central, and Central. Noah has a better performance in the Panhandle and a poorer performance in the Northeast. SMAP has a better performance in the Panhandle, Southwest, and West Central, and is worse in the Northeast, East Central, and Southeast. All three products show their worst performances in the Northeast Oklahoma division.

During the fall season (Figure 7 and Table 7), among all three, Mesonet has better performance in the West Central, Southwest, Panhandle, and South Central and worse performance in the Northeast and East Central. On the other hand, Noah shows better performance in the Panhandle but a poorer performance in the Northeast. SMAP has better performance in the Panhandle, Southwest, West Central, and South Central, and worse performance in the Northeast, East Central, and Southeast. All three products showed the worst performance in the Northeast.

During winter (Figure 8 and Table 8), among the nine climate divisions, Mesonet has higher mean CC values than Noah, while Noah provides lower averaged RMSE values than Mesonet. SMAP values are consistently lower -than-Mesonet (or Noah) averaged CC and higher-than-Mesonet (or Noah) averaged RMSE values for all nine climate divisions. For each product, Mesonet shows the worst performance in the central climate region. Noah has better performance in the panhandle and southwest, which worsens in the central and north-central areas. SMAP has better performance in the southwest and the panhandle but worse in the northeast division.

*4.3. Soil Moisture Product Intercomparison by Land Cover Types*

The TC intercomparison results for Mesonet, Noah, and SMAP over different land cover types are shown in Figure 9 (period-integrated), Figure 10 (CC, across seasons), and Figure 11 (RMSE, across seasons). Since the results for the 6:00 p.m. case are highly similar to those for the 6:00 a.m. case, only the 6 a.m. case is presented.

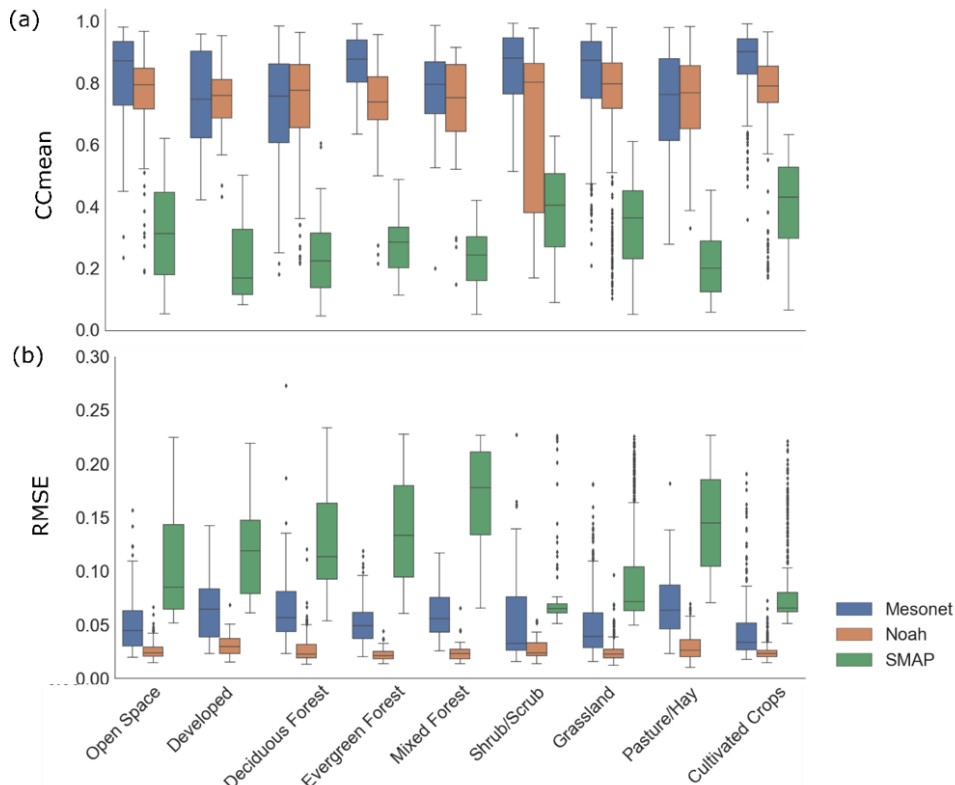

**Figure 9.** Soil moisture product intercomparison results using the TC method for Mesonet, Noah, and SMAP based on (**a**) CC and (**b**) RMSE for values at local 6 a.m., for nine Oklahoma state representative land cover types (see Table 3) from April 2015 through July 2019.

The quartile distribution and data range of the CC and RMSE for each soil moisture product over different land cover types are shown in Figure 9 through box-and-whisker diagrams. Figure 9a illustrates that, in terms of CC, Mesonet shows the highest correlations with the unknown truth in evergreen forest, cultivated crops, and shrub/scrub land cover types, but Noah has slightly better CC values (compared to Mesonet) in deciduous forest and pasture/hay land cover types. Interquartile variability and ranges are similar across categories for Mesonet and Noah, except for the shrub/scrub class, where Noah shows significant, below-the-mean, variability and perhaps outliers. SMAP consistently provides the lowest correlations with the unknown truth in all land cover types. Additionally, according to the RMSE (Figure 9b), Noah provides the lowest mean values and interquartile ranges, followed by Mesonet in all land cover types. For this metric, SMAP appears with the largest values, although its values are lower than Mesonet and Noah's interquartile variability in shrub/scrub and lower than Mesonet's interquartile range in cultivated crops.

The seasonal box and whiskers distribution of the CC for each soil moisture product for different land cover types are illustrated in Figure 10. Overall, despite summer and fall having similar CC values between Mesonet and Noah across diverse land cover types, Mesonet appears to outperform across products and seasons.

During the spring season (Figure 10a), Mesonet, followed by Noah, has the highest $CC_{mean}$ values for all nine land cover types. SMAP $CC_{mean}$ values are consistently lower than Mesonet (or Noah) for all land cover types. The range of variability of the $CC_{mean}$ for Noah at shrub/scrub is the largest of all products.

During the summer months (Figure 10b), the $CC_{mean}$ values seem to vary less within each land cover type (compared with the spring season) for all products. Mesonet provides the highest correlations with the unknown TC truth in evergreen forest, mixed forest, and shrub/scrub land covers. Noah provides similar correlative distributions but slightly lower-than-Mesonet $CC_{mean}$ values in other land cover types (e.g., open space, grassland, developed, deciduous forest, pasture/hay, and cultivated crops). On the other hand,

SMAP consistently illustrates lower-than-Mesonet (or Noah) $CC_{mean}$ values over all land cover types.

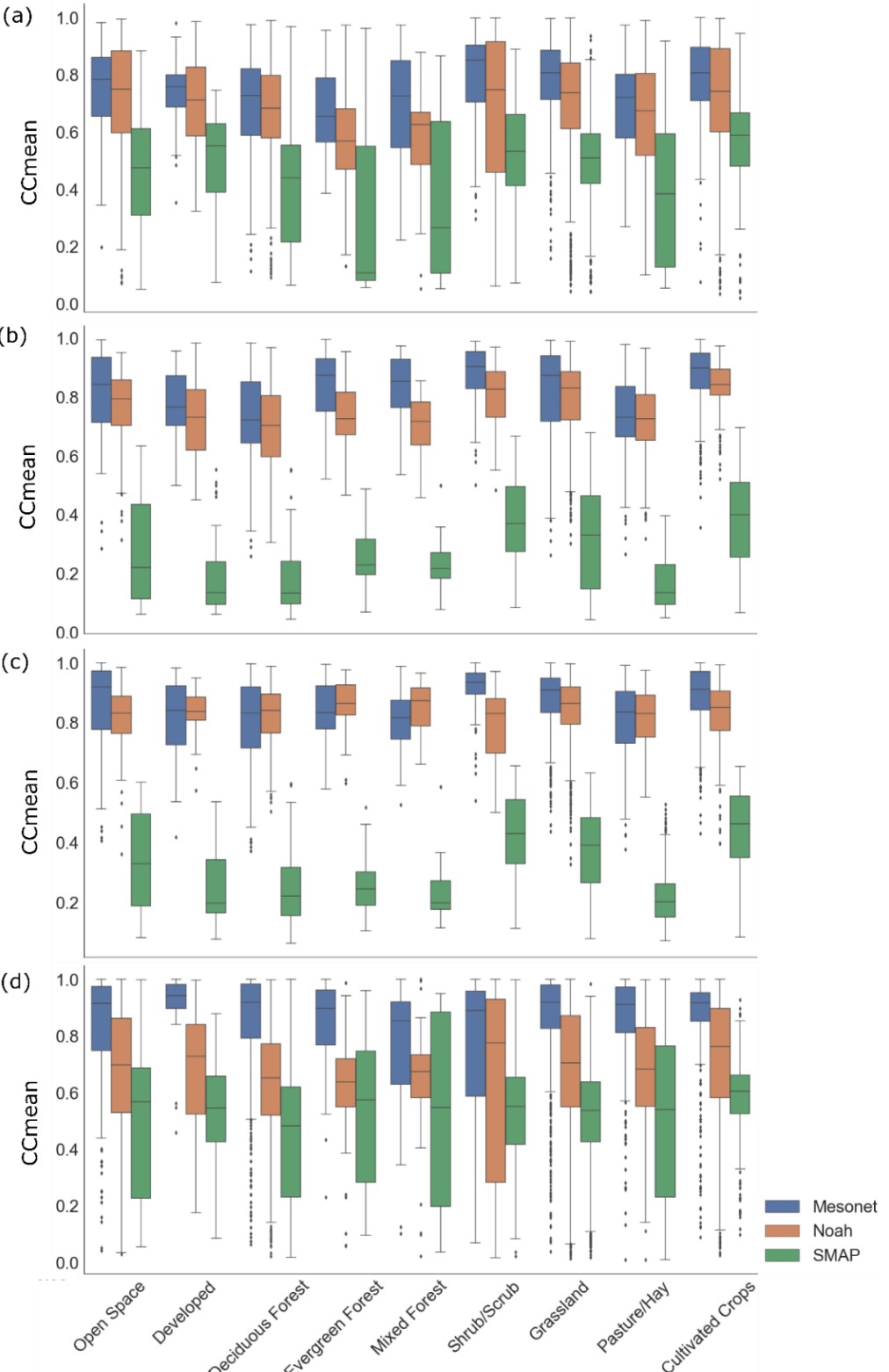

**Figure 10.** Mean correlation coefficient ($CC_{mean}$) between each soil moisture product and the TC-derived "unknown truth" at local 6 a.m., for nine Oklahoma state representative land cover types (see Table 3). Results are discretized by season: (**a**) spring, (**b**) summer, (**c**) fall, and (**d**) winter for the period from April 2015 to July 2019.

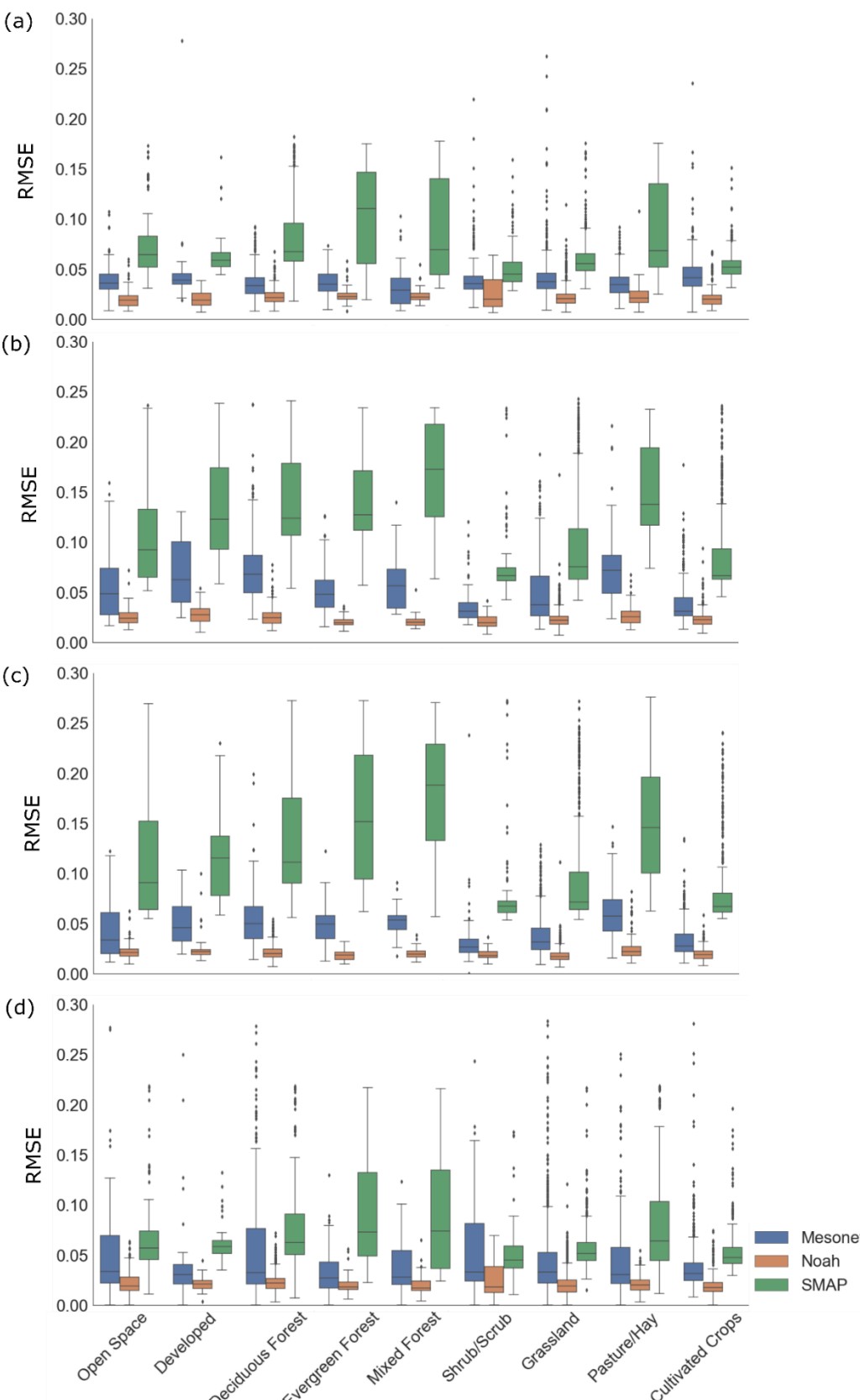

**Figure 11.** Mean RMSE between each soil moisture product and the TC-derived "unknown truth" at local 6 a.m., for nine Oklahoma state representative land cover types (see Table 3). Results are discretized by season: (**a**) spring, (**b**) summer, (**c**) fall, and (**d**) winter for the period from April 2015 to July 2019.

In the fall (Figure 10c), the correlative behaviors are similar to the summer season. Mesonet provides the highest correlation values with the unknown TC truth over shrub/scrub, open space, grassland, pasture/hay, and cultivated crop land cover types. Noah shows the highest $CC_{mean}$ in developed, evergreen forest, deciduous forest, and mixed forest. The $CC_{mean}$ values of SMAP are the lowest over all land cover types. Finally, the winter distributions look similar to the spring conditions, with wider interquartile ranges across products, and Mesonet providing the highest $CC_{mean}$ values in all land cover types.

The seasonal distributions of the RMSE of each soil moisture product and the TC-derived "unknown truth" over different land cover types are shown in Figure 11a–d. Across all seasons, Noah provided the lowest RMSE values for all land cover types. On the other hand, SMAP provided the highest errors, except in winter under open space, shrub/scrub, and grassland land cover types when (and where) the performance was similar to that exhibited by Mesonet.

In summary, the period-integrated TC intercomparison results for Mesonet, Noah, and SMAP for different land cover types (Figure 9) indicated that Noah provided the best performance (with the highest averaged CC and lowest averaged RMSE values) over deciduous forest, and pasture/hay land cover types. For the other land cover types, Mesonet had the highest mean CC values, while Noah provided the lowest averaged RMSE values. SMAP exhibited the poorest performance over all land cover types. The seasonal TC intercomparison results for Mesonet, Noah, and SMAP over different land cover types (Figures 10 and 11) indicated that in the fall, Noah provided the best performance (with the highest averaged CC and lowest averaged RMSE values) and development, evergreen forest, deciduous forest, and mixed forest land cover types. In the spring, summer, and winter seasons, Mesonet reaches the highest mean CC values, while Noah provides the lowest averaged RMSE values over all land cover types. SMAP exhibits the least desirable performance over all land cover types across all seasons. In addition, the larger interquartile ranges presented by Noah for the period-integrated assessment of shrub/scrub (Figure 9a) seem to be induced by large correlation variability during the spring and winter seasons (Figure 10a,d). On the other hand, SMAP illustrates lower-than-Mesonet (or Noah) $CC_{mean}$ values, especially during the spring, summer, and fall, which explain the low period-integrated values.

## 5. Discussion

The TC is a measurement assessment method for estimating the random error variances of three spatially and temporally collocated sampling systems of the same geophysical variable without treating any one system as perfectly observed "truth" [52]. This method has been widely used in the validation of both satellite-based and model-output variables in recent years [29–32]. Despite the fact that the TC has been advancing, several knowledge gaps still exist in relation to its application in soil moisture measurements, including: (1) the lack of understanding of in situ soil moisture product representativity and seasonal performance variability; and (2) the influence of different land cover types on the data quality or simulation skill of each product. This article addresses these two knowledge gaps by conducting a comprehensive assessment of the satellite SMAP L3_SM_P_E, land surface NLDAS_NOAH0125_H, and Oklahoma Mesonet soil moisture products across the state of Oklahoma at daily and seasonal timescales using the TC method evaluated over different land cover types during more than four consecutive years of simultaneous measurements.

The period-integrated TC intercomparison results for Mesonet, Noah, and SMAP over nine Oklahoma state climate divisions (Figure 4 and Table 4) indicate that Noah provided the best performance in the central, northeast, and east-central climate divisions of the state. The same pattern was found in the seasonal TC intercomparison results during the fall season (Figures 5–8). This suggests that it might be inappropriate to regard interpolated Mesonet measurements as the benchmark in the central, northeast, and east-central regions of Oklahoma. The reasons why Noah shows better quality and representativeness in these

regions could be due to: (1) the Oklahoma Mesonet site standards require the sites to be far away from urban landscapes, irrigation, forests, bare soil, fast-growing vegetation, and large bodies of water to minimize those influences [33,34]. (2) The majority of land cover types in these climate divisions are grassland with urban landscapes, pasture/hay, and deciduous forest (Figure 2), where Noah performs better. (3) The Mesonet product used in our TC intercomparison was interpolated from point measurements to match the spatial resolution of SMAP (9 km) using an ordinary kriging method [21,40]. This interpolation method did not consider auxiliary variables, including soil type, land cover, and topography, that affect the true variation of surface soil moisture. Therefore, our interpolated Mesonet product might not be able to well represent the true soil moisture geographical variations in the central, northeast, and east-central regions.

The seasonal TC intercomparison results for Mesonet, Noah, and SMAP over nine climate divisions (Tables 5–8) indicate that in spring and winter, Mesonet has higher mean CC values than Noah, while Noah provides lower averaged RMSE values than Mesonet. According to McColl et al. [28], this suggests that, while the interpolated Mesonet estimates of true soil moisture are noisier than those of Noah (making the Mesonet's RMSE with the unknown truth slightly higher), the interpolated Mesonet has higher Pearson correlation coefficients with the unknown truth, a quantity that is proportional to the unbiased signal-to-noise ratio of Mesonet in the context of the TC method.

Both the period-integrated TC intercomparison and the seasonal TC intercomparison results show that SMAP exhibits the third-highest performance over all climate divisions across all seasons. The reasons for SMAP's lowest performance could be due to: (1) Microwave remote sensing that is responsive to a surface (~5 cm) soil moisture in regions (as opposed to the 10 cm Mesonet and 0–10 cm integrated Noah sample depth) with sparse to moderate vegetation density [46,53–55]. Additionally, the wetter the soil, the shorter the soil sample depth, as the L-band microwave penetration appears to be affected by water content [55,56]. (2) There are challenges with retrievals in areas with complex topography, dense vegetation, near water bodies, or cities [57,58]. Finally, the stripe patterns shown in the $CC_{mean}$ and RMSE maps for the SMAP assessment were found to be an artifact of the SMAP product gridding, possibly reinforced by the contrasting signal attenuation given by the strong west-to-east vegetation density gradient from pastures and sparse trees to dense forests [46,54,55]. On a geographical basis, similar patterns of lower (higher) CC and RMSE are shared among the three products, meaning that their performance decreases (increases) in tandem with the TC's unknown truth. Therefore, despite the relatively low performance of SMAP, we think its inclusion in a multisensory blend is beneficial due to the spatially consistent correlation structures presented with the other two independent products.

In terms of their performance over different land cover types (Figures 9–11), Mesonet provided the best estimates for volumetric soil moisture over shrub/scrub, grassland, and cultivated crops, because the Oklahoma Mesonet site standards minimize the influence of urban landscapes, irrigation, forests, bare soil, fast-growing vegetation, and large bodies of water [33]. It is suggested that vegetation at the Mesonet sites should be uniform and low-growing, such as short grasses [34]. Noah provided the best estimates of volumetric soil moisture over hay/pasture, deciduous forest, mixed forest, and evergreen forest.

Although we conducted and analyzed both the 6 a.m. and 6 p.m. results, only those corresponding to the 6 a.m. time stamp are shown in this study due to the high similarity of the comparative maps of the $CC_{mean}$ and RMSE. This effect obeys the combination of three factors: (1) consistent estimations without significant differences for the same day between 6 a.m. and 6 p.m. individually across systems (i.e., Mesonet, Noah, and SMAP), (2) temporal persistence of the surface soil moisture values across hours, making correlations and similar errors between 6 a.m. and 6 p.m., and (3) interstorm periods, including 6 a.m. and 6 p.m., being more frequent than storm periods and therefore, producing similar results for 6 a.m. and 6 p.m.

Some limitations of this study are: (1) the use of a low number of years (approximately four, dictated by the availability of SMAP) that might not statistically represent the inter-

annual climatic variability of the study region and, therefore, its effects on the estimation of soil moisture, (2) the inherent, systematic, and perhaps correlated (nonrandom) errors across the different measuring platforms, and (3) the fact that the adopted Mesonet product was interpolated from point-scale ground stations to a spatial resolution of SMAP (9 km) using an ordinary kriging method. To minimize the possible negative effect of these limitations, future work could use more years of analysis, conduct a statistical independence test (outside the TC), and interpolate Mesonet with regression kriging approaches, including independent predictors, such as soil properties, land cover, topography, and precipitation to increase the accuracy of the interpolated product. Overall, this study provides a stepping stone for merging these three independent products by acknowledging that spatial representativity is important when wrongly assuming that a point-scale measurement can be up-scaled to a pixel-scale estimation and that different land cover types are critical drivers of soil moisture variability that entail the blend of multisensor products as opposed to a one-size-fits-all approach.

## 6. Conclusions

The objective of this study was to cross-evaluate the accuracy and error characteristics of the most commonly used, yet operationally independent, satellite, model-based, and in situ soil moisture products over the state of Oklahoma. Specifically, the assessment of the SMAP L3_SM_P_E (i.e., SMAP), NLDAS_NOAH0125_H (i.e., Noah), and interpolated Oklahoma Mesonet (i.e., Mesonet) soil moisture products at daily and seasonal timescales was conducted using the triple collocation method, and their performances were evaluated over different land cover types. Several conclusions are summarized as follows:

1.  At the daily timescale, the interpolated Oklahoma Mesonet and Noah were found to be more reliable than SMAP for all considered metrics. Specifically, Mesonet provided the best estimates of volumetric soil moisture with a mean Pearson correlation coefficient of 0.805, followed by Noah with a mean Pearson coefficient of 0.747. However, Noah represents the true soil moisture variation better than our interpolated Mesonet product at a mesoscale with an averaged RMSE of 0.026 $m^3/m^3$. The period-integrated TC intercomparison results for Mesonet, Noah, and SMAP over nine climate divisions indicate that Noah provided the best performance in the central, northeast, and east-central regions.

2.  At disaggregated seasonal timescales, the interpolated Oklahoma Mesonet and Noah were found to be more reliable than SMAP for all metrics in all four seasons. Specifically, Mesonet provided the best estimates of volumetric soil moisture with average correlation coefficients of 0.753, 0.807, 0.855, and 0.811 in spring, summer, fall, and winter, respectively. However, Noah provided the best performance in representing the true soil moisture variation, with an average RMSE of 0.0229, 0.0244, 0.0204, and 0.0217 $m^3/m^3$ in each season, respectively.

3.  In terms of their performance over different land cover types, Mesonet provided the best estimates of volumetric soil moisture over shrub/scrub, grassland, and cultivated crops, but Noah provided the best estimates of volumetric soil moisture over hay/pasture, deciduous forest, mixed forest, and evergreen forest. This reflects the fact that Oklahoma Mesonet site standards minimize the influence of urban landscapes, irrigation, forests, bare soil, fast-growing vegetation, and large bodies of water [33].

4.  Despite the relatively low performance of SMAP in terms of its Pearson correlation coefficient and mean squared errors, the relatively consistent geographic patterns with the unknown truth, reflected by the spatially distributed maps shown in this study, reflect its value in a possible product merger as an independent measurement system of the surface soil moisture.

The TC method-based results of this study provided a new perspective for comparatively assessing multisource soil moisture products and a basis for objective data merging

to capitalize on the strengths of the multisensor soil moisture products in the state of Oklahoma and beyond.

**Author Contributions:** Conceptualization, Z.H., H.A.M., and Y.H.; Methodology, Z.H. and H.A.M.; Software, Z.H., Z.L., and S.L.; Formal Analysis, Z.H.; Writing and Original Preparation, Z.H. and H.A.M.; Writing—Review and Editing, H.A.M., L.V.A., and J.S.G.; Visualization, Z.H. All authors have read and agreed to the published version of the manuscript.

**Funding:** This publication was supported through NOAA Educational Partnership Program/Minority-Serving Institutions awards number NA16SEC4810008 and NA22SEC4810016 to the Center for Earth System Sciences and Remote Sensing Technologies. Contents are solely the responsibility of the author(s) and may not represent official views of NOAA or the U.S. Department of Commerce.

**Data Availability Statement:** The data presented in this study are available on request from the corresponding author. The data are not publicly available at the moment due to Federal data release policies.

**Conflicts of Interest:** The authors declare no conflict of interest.

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
