# Peer review of "Triple Collocation of Ground-, Satellite- and Land Surface Model-Based Surface Soil Moisture Products in Oklahoma—Part I: Individual Product Assessment"

_remotesensing, doi:10.3390/rs14225641_

Round 1

Reviewer 1 Report (Previous Reviewer 3)

The second version is improved especially on the Introduction. It was easy to follow. I still felt the discussion and the conclusion section need some work. There is not much evidence provided to support their work, and some of the conclusion was obvious statement. 

I could not provide comments on the papers, it seems it was turned off. I have attached sticky notes in the attachment for your reference.

Author Response

Please see attached PDF with responses to reviewers.

Reviewer 2 Report (New Reviewer)

This paper examined the characteristics of differences in the surface soil moisture values acquired in the state of Oklahoma using the triple collocation method. In this paper, in situ values from the Oklahoma mesonet are worth being mutually compared to the LSM values from NLDAS-NOAH products and the satellite values from SMAP L3, hence, the triple collocation is best for assessing accuracy of the individual product. The results show the mesonet and the NLDAS-NOAH products are more reliable than the SMAP accompanied by the secondary seasonal characteristics. I think the material is suitable to this journal and the text is well written.
However, I think the discussion and the conclusions can be dug in deeper in terms of some points such as the large errors of SMAP soil moisture and the seasonal change of correlation coefficients. The authors are encouraged to revise adequately for readers' better understanding and clarifying the physical meanings of the results.

Table 2: The sum of AM and PM should be same as the sum of the four seasons.

Figures 4-8: How do the stripe patterns appear?

L378 and etc.: The 6PM results are always highly similar to the 6AM results in every aspect described in this paper. Why are the characteristics of morning and evening results as likely as each other?

L569: This description is followed from that appeared in McColl et al. (2014). I cannot find what this sentence means in the context of this paper.

L571: The poor performance of SMAP is attributed to the two points that have been generally used for explanation in many literature. Recently, the representative depth of soil moisture retrieved by the microwave radiometers such as SMAP and AMSR2 has been discussed deeper.

L576: I see the SMAP performance is relatively good in the three regions but the Mesonet and the Noah performances in the same regions are not significantly bad.

Author Response

Please see attached PDF with responses to reviewers.

Reviewer 3 Report (New Reviewer)

The authors present an interesting first part of two studies on the cross-evaluation of three different spatial resolution surface soil moisture products for different landscapes and climatic regions within Oklahoma state.

The paper has been substantially improved and is almost ready for publication.

In my opinion, the main critical point of the proposed methodology could be the further data merging of these different sensors, platforms and interpolations used to estimate such a difficult variable which shows a high variability even within an hour, i.e. surface soil moisture.

Moreover, Discussions must be improved with the main strengths and limitations of their work better highlighted, this also in view of the second part of this study.

Minor comments:

Figures 4, 5, 6, 7, and 8: please add the X and Y captions on the six figures and improve the quality and readiness of the figures. They are not very clear to me.

Author Response

Please see attached PDF document with responses to reviewers

Reviewer 4 Report (New Reviewer)

This paper examined a TC approach to the intercomparison of the three soil moisture data sets, which is interesting but seems to need a change of research design for the original purpose of the TC method.

Line 331. RMSE is not a correct expression. RMSD is correct because it is about the difference between the three data sets, not the error.

Tables 4 to 8. SMAP had quite different values from the other two data sets. It is not sure why the SMAP dataset is necessary. It seems to amplify the uncertainty of the data merge.

Line 593. I do not think the TC approach is appropriate for merging in-situ (Mesonet) and estimated (SMAP and Noah) values. Indeed, the in-situ soil moisture is the correct answer. The estimated soil moisture values by a satellite (SMAP) and a model (Noah) should be evaluated using the right answer. In-situ is a true value, while estimation is just a guess. TC approach is appropriate for merging three data sets whose true values are not easily found. Alternatively, the authors can use the TC for SMAP, Noah, and ERA5 soil moisture. The accuracy of SMAP, Noah, and ERA5, respectively, may be improved by merging SMAP, Noah, and ERA5, regarding the true value (Mesonet).

Author Response

Please see attached pdf document with responses to reviewers

Round 2

Reviewer 2 Report (New Reviewer)

I think the revised version is fairly improved from the older version and worth being published.

This manuscript is a resubmission of an earlier submission. The following is a list of the peer review reports and author responses from that submission.

Round 1

Reviewer 1 Report

The study by Hong et al., has cross evaluated three soil moisture datasets through the Triple Collocation method across Oklahoma state of USA. This study has analyzed the impact of land-cover, and also seasonal changes on the accuracy of soil moisture products. While the in-situ measurements are also distributed to grid level as another source of soil moisture measurements. This study finds that overall, the in-situ measurements and Noah soil moisture products are reliable sources of soil moisture observations, while these performances vary spatiotemporally. Although this study shows a lot of experiments and results, overall, lacks from motivation and the innovation of the study is un-clear. The high accuracy of in-situ soil moisture products and Noah simulations are already shown in past literature both over different land-covers, and also different seasons/time periods. Hence, I suggest authors to work more on the innovation part of their study more, as the moment, I don't see enough contribution to their work to encourage it for publications.

Reviewer 2 Report

There is a technical editor error where the text does not show up after equation 4. 

The tables I reference intermingle results for correlation coefficient and root mean square error. Consider separating these into separate columns.

Otherwise, I felt it was a good paper not in need of any substantial revision and should be accepted in its present form, with the above caveats. 

Reviewer 3 Report

Overall, this paper is lengthy, hard to follow, and unorganized. I know the authors put tremendous effort in getting all the data and analyzing it, but with weak storytelling, it's hard to relate of what the gap and what is the significance of this work? I think this could be a useful paper but the manuscript will need a major revision before it is publishable. Please work on rewriting the section. The results section needs to be trimmed and the discussion section needs a major revision. I have attached a few edits in the attached pdf. 
